# In Sickness and in Health: The Immunological Roles of the Lymphatic System

**DOI:** 10.3390/ijms22094458

**Published:** 2021-04-24

**Authors:** Louise A. Johnson

**Affiliations:** MRC Human Immunology Unit, MRC Weatherall Institute of Molecular Medicine, University of Oxford, John Radcliffe Hospital, Headington, Oxford OX3 9DS, UK; louise.johnson@imm.ox.ac.uk

**Keywords:** lymphatic, dendritic cell, inflammation, migration, lymph node, T-cell, trafficking, high endothelial venule, adhesion molecule, chemokine

## Abstract

The lymphatic system plays crucial roles in immunity far beyond those of simply providing conduits for leukocytes and antigens in lymph fluid. Endothelial cells within this vasculature are distinct and highly specialized to perform roles based upon their location. Afferent lymphatic capillaries have unique intercellular junctions for efficient uptake of fluid and macromolecules, while expressing chemotactic and adhesion molecules that permit selective trafficking of specific immune cell subsets. Moreover, in response to events within peripheral tissue such as inflammation or infection, soluble factors from lymphatic endothelial cells exert “remote control” to modulate leukocyte migration across high endothelial venules from the blood to lymph nodes draining the tissue. These immune hubs are highly organized and perfectly arrayed to survey antigens from peripheral tissue while optimizing encounters between antigen-presenting cells and cognate lymphocytes. Furthermore, subsets of lymphatic endothelial cells exhibit differences in gene expression relating to specific functions and locality within the lymph node, facilitating both innate and acquired immune responses through antigen presentation, lymph node remodeling and regulation of leukocyte entry and exit. This review details the immune cell subsets in afferent and efferent lymph, and explores the mechanisms by which endothelial cells of the lymphatic system regulate such trafficking, for immune surveillance and tolerance during steady-state conditions, and in response to infection, acute and chronic inflammation, and subsequent resolution.

## 1. Introduction

Immunology is at the cutting edge of medical research, and the exponential increase in knowledge of this rapidly developing field has led to better understanding and improved treatment of infectious diseases, autoimmune disorders and cancer. However, the vital roles that the lymphatic system plays in this exciting field are frequently overlooked, from normal day-to-day immune surveillance to aiding immunological responses to pathogenic challenges as they arise, and resolving subsequent inflammation.

The lymphatic system comprises lymphatic vessels, organs that contain lymphoid tissue such as the spleen, lymph nodes, Peyer’s patches and thymus, and the lymph fluid that flows throughout. Unlike the blood system, the lymphatics are not a circulatory network: instead, cells and fluid move through in one direction (Figure 1A). Excess interstitial fluid, macromolecules and leukocytes exit the tissue via blind-ended afferent lymphatic vessels and enter draining lymph nodes via the subcapsular sinus. Lymph nodes are critical for immune surveillance, providing a highly organized hub in which blood-derived naïve lymphocytes might encounter antigen and antigen-presenting cells borne in the afferent lymph. After percolating through the lymph node, fluid and lymphocytes return to the blood circulation through efferent lymphatic vessels and the thoracic duct.

In the early 1950s, all that was known about “lymphocytes” was that they were small, round, motile cells that were found in lymphatic vessels and thus acquired their name. The pioneering work of Sir James Gowans and colleagues demonstrated that lymphocytes were not newly formed in lymph nodes as was initially assumed but in fact recirculating [1,2,3]. Gowans’ elegant experiments showed that following thoracic duct cannulation in rats, there was a progressive loss of lymphocytes in the thoracic duct lymph. This could be reversed by returning these collected cells to the rats by intravenous injections, whereupon such cells could later be recovered in lymph from the thoracic duct.

Later investigators continued to examine the traffic of lymphocytes through individual tissues in larger animals, typically sheep. Ovine models are particularly useful for studying lymphocyte traffic, as (in addition to ease of surgical access to individual lymph nodes) there is usually only a single efferent lymphatic vessel, draining each lymph node directly to the thoracic duct [4]. In humans and rodents, the “plumbing” is more complex, with lymph nodes often occurring in chains whereby the efferent lymph of one node contributes to the afferent supply of a second node. Experiments in sheep demonstrated preferential recirculation of distinct pools of lymphocytes, either through subcutaneous lymph nodes, mesenteric lymph nodes or through skin [5,6]. It also became apparent that there were differences in the cellular composition of afferent lymph, efferent lymph and blood [4,7,8]. As with many areas of research in biological sciences, our understanding improved massively upon the advent of monoclonal antibodies, which permitted more thorough characterization of lymphocyte subsets, as well as other leukocytes such as dendritic cells (DCs), macrophages and neutrophils. Clearly, there is selectivity as to which leukocytes are permitted to enter the lymphatic system and this review details the molecular regulation of this, particularly that imposed by endothelial cells of high endothelial venules, lymphatic vessels and lymph node sinuses.

## 2. In a Healthy Steady-State

The immune system of a non-inflamed, non-infected individual may be described as “resting” but it is by no means static, with cell migration a constant feature throughout the blood and lymphatic networks. Studies on rodents, sheep and humans have revealed that afferent lymph draining normal healthy skin typically contains approximately 10-15% macrophages or DCs whereas efferent lymph contains only lymphocytes. Meanwhile, around 40% peripheral blood lymphocytes (mostly B-cells) are non-circulating cells and thus do not enter lymphoid tissue (Figure 1B), [4,9,10,11,12].

### 2.1. Entering Lymph Nodes from the Blood

Blood-borne lymphocytes enter the cortex of lymph nodes through specialized post-capillary venules termed high endothelial venules (HEVs), found in all secondary lymphoid organs with the exception of the spleen (reviewed in [13]. The high endothelial cells (HECs) of this vasculature are easily distinguishable from other endothelial cells by their raised, cuboidal morphology. In addition to expressing pan-endothelial cell markers such as CD31 and vascular endothelial cadherin (VE-cadherin) [14], HECs have a unique transcriptional profile to permit selective lymphocyte recruitment [15] in a multi-step adhesion cascade, involving tethering, rolling, adhesion and transmigration [16,17,18,19,20]. The cuboidal structure of HECs results in a more irregular lining of these venules, increasing turbulence in blood flow. Consequently, HECs have greater adhesiveness for circulating lymphocytes and a collision with the vessel wall will result in a loose attachment which may last several seconds [18,21,22]. A similar collision occurring with normal blood vessel endothelial cells of the microcirculation would cause the lymphocyte to immediately rebound or adhere very briefly before being swept on by the force of the blood flow.

Initial tethering of lymphocytes is supported by 6-sulpho sialyl Lewis^X^ motifs decorating O-glycans and N-glycans of sialomucins, a family of sulphated, fucosylated and sialylated glycoproteins termed peripheral node addressins (PNADs), which are displayed on HECs and bind the lymphocyte homing receptor L-selectin (CD62L). Rolling lymphocytes are then activated, in part through the sheer force of blood flow but also through G protein-coupled chemokine receptors [23]. CC-chemokine ligand 21 (CCL21), immobilized on the luminal HEC surface by heparan sulphate, binds CC-chemokine receptor 7 (CCR7) on both naïve T-cells and B-cells [24,25,26,27,28], as well as on plasmacytoid DCs (pDCs) [29], The CXC-chemokine receptor CXCR4 also contributes to lymph node homing through binding CXCL12, which is broadly expressed on HEVs [30]. A further chemokine–receptor pair, CXCL13 and CXCR5, has been shown to regulate B-cell entry to both lymph nodes and to Peyer’s patches, lymphoid nodules in the small intestine [30,31].

Curiously, chemokines are not always simply synthesized by HECs and secreted or presented apically. Although CCL21 is expressed in HEVs of murine lymph nodes, human HEVs lack CCL21 mRNA, and this chemokine, like CCL19, CXCL12 and CXCL13, is produced by stromal cells within the lymph node [15,32,33,34,35]. Additionally, chemokines may be transported in afferent lymph from peripheral tissues [36], particularly during inflammation (discussed later in this review). Subsequently, chemokines are bound at the basolateral surface of HECs, internalized and then presented on the apical surface following transcytosis by molecules such as the atypical chemokine receptor ACKR1 [37].

The “inside-out” signaling induced by ligation of chemokine receptors with their cognate ligands in the tethered lymphocyte induces conformational changes of the cell adhesion molecule LFA-1 (αLβ1 integrin) [38,39]. Activated integrin LFA-1 on naïve T- and B-cells binds ICAM-1 and ICAM-2 on the endothelium, mediating firmer binding. There is also some contribution of VCAM-1, particularly in the smallest HEVs, through interacting with VLA-4 (α4β7 integrin) [40] following rapid (< 0.1s) chemokine-induced clustering of VLA-4 within lymphocyte-endothelial contact zones [41]. Additionally, in mesenteric lymph nodes, VLA-4^+^ lymphocytes engage with the mucosal addressin adhesion molecule MAdCAM-1 [19,20,38].

Integrin activation of HEV-homing T-cells is further amplified by the bioactive lipid mediator sphingosine-1-phosphate (S1P), triggering signaling through its G-protein coupled receptor S1P_1_. This endogenous sphingolipid is produced by sphingosine kinases Sphk1 and Sphk2, expressed in most eukaryotic cells [42], with lymph node lymphatic endothelial cells (LECs) providing a major source of S1P [43]. A study using S1P_1_^−/−^ CD4^+^ T-cells showed that such cells rolled equally well as S1P_1_^+^ T-cells but exhibited reduced firmer adherence, suggesting that S1P_1_ is required for optimal integrin activation [44].

After crawling on the HEC surface for several minutes, lymphocytes rapidly transmigrate across the endothelium and crawl along the highly organized stromal networks of the lymph node, guided by stromal-derived chemokines [45,46,47]. For additional information on intranodal positioning of leukocytes, the reader is referred to [48]. In brief, T-cells localize to T-cell-rich areas of the paracortex, guided by the chemokines CCL21 and CCL19, whereas B-cells enter B-cell follicles in the cortex, through CXCL13 as well as CCL21 and CCL19. Here, these lymphocytes await possible encounters with their cognate antigens, to be delivered to them from the periphery via afferent lymphatic vessels.

### 2.2. Egress from Peripheral Tissues via Afferent Lymphatics

In peripheral tissues, blind-ended capillaries of afferent lymphatic vessels form an extensive network**,** removing excess fluid that has leaked out from the blood vasculature into the interstitium and returning it to the blood circulation, but only after thoroughly immune surveillance in downstream lymph nodes. In addition to soluble molecules, lymph carries leukocytes: predominantly T-cells (80–90%) and dendritic cells (DCs), (5–15%), with smaller numbers of monocytes, macrophages, B-cells and granulocytes [8,49], (Figure 1B). The majority of afferent lymph-borne T-cells are CD4^+^ effector-memory T-cells, with naïve T-cells representing only a minor subset [50,51].

In the steady state, continuous recirculation of memory T-cells through peripheral tissues facilitates enhanced immunosurveillance and rapid responses to reinfection [50,52,53,54], while egress of DCs from tissues is essential for maintaining peripheral tolerance [55]. DCs capture and process soluble molecules, either self-antigens (for example, those derived from dying cells) or non-self-antigens such as harmless environmental molecules, for presentation via MHC class I and class II molecules. However, DC maturation is not triggered as such antigens are at a low dose and there is an absence of “danger” signals [56]. Therefore, when such antigen is presented to the corresponding T-cells by these regulatory or tolerogenic DCs in draining lymph nodes, the autoreactive T-cells are deleted and subsequently unresponsive to future similar antigenic challenges.

#### 2.2.1. Functionally Specialized Architecture of Lymphatic Capillaries

Most leukocytes enter the proximal half of initial lymphatic capillaries, which are also the likely sites at which fluid accesses the afferent lymphatic system [57,58,59]. To aid fluid drainage, these blind-ended capillaries lack a continuous basement membrane or smooth muscle cells, unlike larger collecting vessels (Figure 2). The endothelial cells themselves have a highly distinctive oakleaf shape that permits them to interdigitate and form loose discontinuous junctions [60]. Neighboring cells are “buttoned” together with the adherens junction protein vascular endothelial cadherin (VE-cadherin) and tight junction proteins occludin, endothelial cell selective adhesion molecule (ESAM), junctional adhesion molecule-A (JAM-A), zonula occludens-1 (ZO-1) and claudin-5. CD31 and lymphatic vessel endothelial receptor for hyaluronan (LYVE-1) are expressed on the flaps in between, which form openings (0.5–1 µm) through which migrating leukocytes can squeeze. These button-like junctions contrast with the continuous “zipper-like” junctions in the collecting lymphatic vessels and blood vessels, which are believed to function more as conduits rather than entry sites.

#### 2.2.2. Chemotactic Guidance

As with leukocyte recruitment via HEVs, CCL21 and CCR7 are crucial for egress of both T-cells and DCs from peripheral tissue [24,61,62,63,64,65,66], Table 1, whereas CCL19 is dispensable [67,68]. CCR7^−^ effector T-cells remain resident within peripheral tissues as sentinels, in contrast to CCR7^+^ recirculating memory T-cells and immunosuppressive CD4^+^ regulatory subsets [69,70]. Curiously, both CD8^+^ and CD4^+^ T-cells express CCR7 and yet CD4^+^ cells exit skin more efficiently than CD8^+^ cells [61,62,71,72]. It is possible that CCR7 function may be modulated by signals mediated by other receptors, as has been described in DCs [73]. Additionally, as T-cells are not completely absent from the lymph nodes of CCR7^−/−^ mice [65], compensatory mechanisms must exit, either to mediate cell entry from the blood or from peripheral tissue.

CCL21 is expressed constitutively by LECs [74,75], secreted, and immobilized to heparan sulphates, thus forming a steeply decaying perilymphatic gradient which guides CCR7^+^ DCs towards lymphatic capillaries from distances of up to 90 µm [68]. Somewhat surprisingly, LEC-produced heparan sulphates are dispensable for the formation of the interstitial CCL21 gradient [76]. Following specific genetic abrogation of heparan sulphate production by lymphatic endothelium, there is a modest reduction in levels of CCL21 at the lymphatic capillary but the CCL21 gradient anchored to mesenchymal heparan sulphates remains intact. Subsequent direct contact between DCs and LECs stimulates endothelial calcium fluxes that in turn trigger local release of CCL21 from intracellular depots within the trans-Golgi network and intracellular vesicles [77]. Such acute chemokine release might attract a migrating leukocyte to enter the lymphatic vessel via a specific entry portal [78], where physical pushing is sufficient to open the button-like junction between LECs [60], with successive migratory cells using the same pre-formed portal in an integrin-independent manner [79,80]. Entry of DCs is also guided by the LEC-expressed GPI-anchored protein semaphorin 3A, through stimulating actomyosin contraction within the DC uropod by interacting with the receptor components Plexin A1 and Neuropilin-1, to facilitate squeezing through the button junctions into the lymphatic lumen [81]. Once inside afferent lymphatic capillaries, DCs must actively crawl in a semi-directed manner, again guided by CCL21 gradients [78,82] until reaching collector vessels, whereupon lymph flow is faster and transport is passive.

A second chemotactic pathway regulating egress or retention of T-cells in peripheral tissue is lymphatic endothelial-derived S1P [43] and its receptor, S1P_1_, which is highly expressed on recirculating CD4^+^ T-cells [83]. The immunosuppressive drug FTY720 (also known as fingolimod) is phosphorylated in vivo and acts as an agonist for S1P receptors, leading to downregulation of expression and subsequently suppressing function [84]. Such pharmacological agonism of S1P_1_ has been demonstrated to impair entry of CD4^+^ T-cells into afferent lymphatics [83] while S1P_1_ over-expression in CD8^+^ T-cells prevented retention in intestine, kidney, salivary gland and skin [85].

**Table 1 ijms-22-04458-t001:** Summary of the molecules known to be involved in lymphatic trafficking of leukocytes.

	Homeostasis	Additional Molecules Involved under Inflammatory Conditions
Migration into and through afferent lymphatics	**CCL21:CCR7** [24,61,62,63,64,65,66] *DCs and T-cells* **S1P:S1P_1_** [43,83,84,85] *T-cells* **Semaphorin 3A:Plexin A1 + neuropilin-1** [81] *DCs* **MR:L-selectin** [86] *T-cells* **MR:CD44** [87,88] *T-cells* **CLEVER-1** [89,90] *T-cells*	**LYVE-1:HA:CD44** [91,92,93,94,95] *DCs* **VCAM-1:VLA4** [96,97,98,99] *DCs and neutrophils* **ICAM-1:LFA-1** [96,97,98,99,100,101,102,103] *DCs, T-cells and neutrophils* **ALCAM** [104,105] *DCs* **L1CAM** [106] *DCs* **CD99** [99,107] *DCs and neutrophils* **CD31** [107] *DCs* **CXCL12:CXCR4** [107,108,109,110,111,112,113] *DCs* **CD69** [114] *T-cells* **E-selectin** [99] *Neutrophils* **CXCL8** [99] *Neutrophils* **CX3CL1:CX3CR1** [115,116] *DCs* **ACKR2**, scavenging CCL2 and CCL5 [117,118,119,120] *Antigen-presenting cells and T-cells* **CCL21:CCR7** [121] *Neutrophils (as well as DCs and T-cells)* **ACKR4**, scavenging CCL19 [122] *Antigen-presenting cells* **CCL2:CCR2** [123] *DCs and Langerhans cells* **ROCK** [124] *DCs*
Entry to lymph node via afferent lymphatics and migration within sinuses	**CCL21** and **CCL19**, via **CCR7** [24,25,28,65,125] *DCs and T-cells* **MR:CD44** [88] *T-cells* β1, β2, β7 and/or αv **integrins** [126] *T-cells* **CCL1:CCR8** [127] *DCs* **PLVAP** [128] *T-cells* **ACKR4** and CCR7 ligands [129] *DCs*	**CD209** [130] *Neutrophils*

#### 2.2.3. Adhesion Molecules

In addition to chemotactic cues, certain adhesion molecules have been implicated in regulating entry into the afferent lymphatic capillaries under steady-state conditions (Table 1). Macrophage mannose receptor (MR, CD206), a C-type lectin expressed on tissue macrophages, immature DCs, lymphatic endothelium and lymph node sinuses, has been reported to bind leukocyte-expressed L-selectin [86] and CD44 [87,88], mediating CD4^+^ and CD8^+^ T-cell egress from the skin. Similarly, CLEVER-1 (Common lymphatic endothelial and vascular endothelial receptor-1) has been shown to mediate transmigration of peripheral blood mononuclear cells across cultured lymphatic endothelial cell monolayers, although the leukocyte-expressed binding partners are yet to be defined [89,90]. Integrins and selectins do not appear to be involved in leukocyte entry into afferent lymphatics under steady state conditions [80,83]. It is important to note, however, that the majority of studies on T-cell trafficking have been carried out using adoptively transferred T-cells derived from lymph nodes or spleens of donor mice and may not necessarily behave in a similar manner to those of endogenous recirculating lymphocytes, or truly represent steady-state, non-inflamed conditions.

### 2.3. Entering Lymph Nodes via Afferent Lymphatics

Afferent lymphatics bear lymph fluid, antigens and leukocytes to tissue-draining lymph nodes, accessing these highly ordered structures via the subcapsular sinus (SCS), situated below the surrounding collagen-rich capsule. Lymph nodes provide two levels of filtration: firstly, in the SCS, providing innate immunological surveillance; and secondly in the parenchyma, where adaptive responses may be generated. In the SCS, lymph-borne cells and solutes are sampled by resident innate immune cells such as CD169^+^ macrophages and CD11b^+^ DCs [131], which identify potentially dangerous particles through pattern recognition receptors (PRRs), for retention and processing in the sinus. The second level of filtration is more selective, permitting only low molecular weight particles (<70 kDa) [36] and cells expressing specific chemokine receptors to penetrate into the T- and B-zones of the parenchyma. Experiments whereby activated CD4^+^ T-cells were transferred directly into afferent lymph revealed that such cells are initially retained in the SCS space by a three-dimensional array of cells and fibers that provide a mechanical sieve [126]. Arrested T-cells subsequently crawl randomly on the sinus floor, in a manner independent of either chemokines or cell adhesion molecules. However, in the parenchyma, there was a marked reduction (almost 50%) in numbers of T-cells deficient in the genes encoding β1, β2, β7 and αv integrins, demonstrating that integrins contribute to the translocation of T-cells from the SCS into the parenchyma, potentially enhancing T-cell sensing of chemokine gradients. Indeed, exit of T-cells from the SCS and migration to the T-cell zone of the parenchyma is sensitive to pertussis toxin, showing the importance of chemokine gradients and Gαi-protein coupled receptors in intranodal positioning [126]. Lymph-borne DCs also home to the T-cell-rich paracortex, mediated by CCR7 [65,125] as well as CCR8 and CCL1 [127] in an integrin-independent manner [80] (Table 1).

In addition, PLVAP (plasmalemma vesicle-associated protein, also known as PAL-E or MECA-32 antigen) is expressed on LECs in the lymphatic sinus and further contributes to selective entry of soluble molecules and leukocytes [128]. Curiously, PLVAP also appears to control cellular traffic, with immune cells crossing the sinus floor presumably by transcellular migration. However, the other molecules involved in this have yet to be demonstrated. Of note, expression of PLVAP is much lower in peripheral LECs than those of the SCS [132] and this molecule was originally believed to be specific for blood vessel endothelium [133,134].

#### 2.3.1. Specialized Subsets of Lymphatic Vasculature within the Lymph Node

LECs within the lymph node do not serve merely as conduit lining but rather are responsible for modulating adaptive immune responses, through antigen capture, presentation and archiving, as well as maintaining tolerance [135]. Recent gene profiling has revealed niche-specific functional specialization and specific markers of LECs within the lymph node, core subsets of which are common to both mouse and human [130,136]. These five main subsets include *Foxc2^+^* valve LECs, *Ackr4^+^* SCS ceiling LECs, *Ccl20^+^* SCS floor LECs, Marco-LECs and Ptx3-LECs (Figure 3). There are also two transitional populations: bridge cells, linking ceiling LECs and floor LECS, and transitional zone LECs, between floor LECs and Marco-LECs.

ACKR4^+^ ceiling LECs are most similar to valve cells, expressing extracellular matrix components and proteins to maintain cell-cell contacts [130,136]. ACKR4 itself is an atypical chemokine receptor, playing a vital role in regulating entry of CCR7^+^ DCs to the lymph node by scavenging CCR7 ligands from the sinus lumen and creating functional CCL21 gradients across the SCS floor [129]. Floor LECs appear more immunologically active, expressing transcripts for adhesion molecules (ICAM-1, VCAM-1, Glycam1, MAdCAM1) and chemokines (CCL20, CXCL4, CXCL1) suggestive of roles in leukocyte migration, as well as immune cell function modulators (CXCL4, CSF1, BMP2/SMAD), MHC class II presentation (CD74, H2-Ab1) and tolerance (PDL1) [130,136,137]. Lymph from the SCS then passes through medullary sinuses, coming into contact with Marco-LECs. These cells share gene expression patterns with myeloid cells, which may indicate a role in innate immune functions. MARCO (macrophage receptor with collagenous structure) is a scavenger receptor, also expressed by medullary macrophages and aiding phagocytic clearance of both Gram-positive and Gram-negative bacteria [138,139]. As yet, it is not known whether it performs a similar function in LECs.

Gene expression in Ptx3-LECs is most similar to that of LECs from peripheral tissues (although *Ptx3* itself is not reported to be highly expressed in peripheral LECs) and both share certain morphological features, including blind ends specialized for fluid uptake [60,136,140,141]. Ptx3-LEC express LYVE-1, which is a widely used lymphatic marker [142] and has been shown to promote DC entry to lymphatic vessels in mouse dermis [91], discussed later in this review. Macrophage mannose receptor MR is also expressed both in peripheral LECs and Ptx-LECs, orchestrating lymphocyte egress from tissues through interacting with the leukocyte receptor CD44 [88], as well as binding microbes and promoting phagocytosis [143]. Additionally, expression of *Ccl21* has been detected in Ptx3-LECs [136]. Whether LYVE-1 and MR perform similar functions in Ptx3-LECs as they do during exit from peripheral tissue remains to be demonstrated but it is possible that these molecules and CCL21 facilitate egress of CCR7^+^ naïve B- and T-cells, in addition to S1P [43,144].

The aforementioned Ptx3 is a member of the pentraxin family, which act as soluble PRRs, binding pathogens and damaged self-proteins, promoting phagocytosis and mediating activation of complements [145]. However, Ptx3-LECs of the central medullary and paracortical sinuses may also play roles in stabilizing the extracellular matrix (ECM), as Ptx3 also binds collagen and fibrinogen-domain containing proteins (including ECM components), [146]. Furthermore, Ptx3-LECs display the transcript for IαI (inter-alpha-trypsin inhibitor) heavy chain 5. IαI heavy chain proteins can interact with both Ptx3 and hyaluronan (HA) [147,148,149], and thus it is tempting to speculate that links between LYVE-1, HA, IαI heavy chain 5 and Ptx3 may provide further structural integrity. Another Ptx3-LEC gene, *Reln*, encodes reelin, an ECM glycoprotein that serves as an important regulator of lymphatic vasculature development in the periphery, contributing to cross-talk between collecting lymphatic vessels and smooth muscle cells [150]. It seems likely that reelin signaling is involved during lymph node development too, and potentially during remodeling such as that induced during prolonged inflammatory stimuli.

#### 2.3.2. Tolerance

Although the maintenance of peripheral tolerance is traditionally ascribed to lymphatic-migratory CD8α^+^ DCs that cross-present self-antigen [151], lymph node-resident LECs also play an important role. Such LECs express numerous peripheral tissue antigens and can mediate deletion of specific cognate CD8^+^ T-cells [152]. SCS floor LECs are the most likely of lymph node LECs to be the key players in this, firstly as their location gives them access to lymph-borne tissue-derived antigens and secondly, because single cell analysis has shown that they express high levels of genes suggestive of effective antigen presentation [136]. Both human and mouse LECs express MHC class I and MHC class II molecules [153]. However, in addition to MHC-antigen complexes, effective activation of T-cells requires antigen-independent signals, provided by co-stimulatory molecules such as CD40, CD80 and CD86. These molecules are virtually absent from LECs, even following inflammatory stimuli. Instead, lymph node-resident LECs express inhibitory receptors such as PD-L1 (CD274) and PD-L2 (CD273), which engage counter-receptors on activated CD8^+^ T-cells to dampen the immune response. The reader is directed to reviews by [154] and [135] for further details on the immunomodulatory roles of LECs.

#### 2.3.3. The Fibroblastic Reticular Network

Once leukocytes have migrated through the floor of the SCS, further infiltration into the paracortex is supported by fibroblastic reticular cells (FRCs) [125]. These specialized fibroblasts synthesize and organize fibers of extracellular matrix components, including collagen, forming a conduit network [141,155,156]. This permits lymph fluid flow while providing a scaffold for leukocyte migration. In addition to secreting CCL19 and CCL21, FRCs also express podoplanin [157], a heavily *O*-glycosylated type I protein widely used as a LEC marker in peripheral tissue [74,158,159]. Interactions between podoplanin and the C-type lectin-like receptor 2 (CLEC-2) [160], found on the surface of DCs and up-regulated upon maturation, are critical for efficient transit in the FRC network [161,162]. Additionally, podoplanin has been shown to form a complex with CCL21 that is subsequently shed [163], potentially contributing further to the CCL21 gradient. For more detail on recent advances into our understanding on DC migration through the FRC network, the reader is referred to [164].

#### 2.3.4. Regulation of Lymph Node Function by Afferent Lymph

Afferent lymph also plays a crucial role in ensuring the maintenance of normal lymph node function. Studies in mice have shown that when lymph nodes are deprived of lymph following occlusion of the afferent lymphatics, HEVs become flat walled, luminal expression of PNAd is lost and HEV-specific genes including *Glycam1* are down-regulated [165,166]. Moreover, these modified HEVs support minimal lymphocyte extravasation, while macrophages disappear from the SCS. Subsequent investigations demonstrated a crucial role for lymph-borne CCR7^+^ CD11c^+^ DCs in HEV maintenance, supporting HEV formation and function by secreting VEGF and lymphotoxin, as well as stimulating CCL21 production in FRCs [167,168].

### 2.4. Exiting Lymph Nodes via Efferent Lymphatics

Naïve lymphocytes that do not encounter their cognate antigen migrate through medullary sinuses and the subcapsular region, before exiting the lymph node via efferent lymphatic vessels. Evidence that this egress is non-random but rather controlled originally came from studies by Binns and colleagues, who injected labelled sheep lymphocytes into neonatal pigs and labelled pig lymphocytes into fœtal sheep, and then assessed the route of exit [169,170]. Pig lymphocytes exited via the efferent lymph in the recipient sheep while sheep lymphocytes left via the blood from the porcine lymph nodes. Thus, the stromal cells native to the recipient animal dictated the migratory route taken by lymphocytes. Regulation of this step is still poorly understood, although now it is known that the “choice” between retention or egression of both B- and T-cells is made by G-protein coupled receptors and their ligands. Lymphocyte retention is promoted by Gαi-coupled receptors (primarily CCR7) whereas S1P_1_ signaling overrides this, mediating entry into cortical sinus central branches of the lymph node [17,43,84,171,172,173]. Additionally, adhesion molecules CLEVER-1 and MR are expressed on efferent lymphatic vessels and blockade of these has been shown to reduce T-cell binding to lymph node sinuses in adhesion assays using frozen sections of lymph node [86,90], although these findings have yet to be confirmed in vivo.

In animals (such as humans and rodents) in which lymph nodes occur in chains, naïve T-cells can migrate from one lymph node to another, as an efferent vessel of one lymph node serves as an afferent lymphatic for the next [174], until returning to the blood circulation via the thoracic duct and subclavian veins, into the vena cava. Thus, under normal conditions of homeostasis, some leukocytes recirculate through the body whereas others are resident in both peripheral and lymphoid tissue, so that all are poised to mount a swift immune response when necessary.

## 3. Acute Inflammation and Infection

One of the striking effects of inflammation or infection is the increase in numbers of DCs egressing from the affected tissue and entering afferent lymphatic vessels [175], orchestrated by chemokine receptors and adhesion molecules. The lymphatic system provides another layer of molecular control in the form of chemokines and counter receptors, summarized in (Table 1).

### 3.1. Cell Migration in Peripheral Tissue and Entry into Afferent Lymphatics

#### 3.1.1. DCs and T-Cells

Inflammation is an essential and complex response to biological, chemical or physical stimuli, which must be carefully regulated and targeted, to remove potential threats to the body but without causing excessive damage to healthy tissue [176]. In the acute phase, the infectious agent or foreign material must be cleared, along with any dead or dying cells that have been damaged by the injury. Inflammation-associated cytokines including TNFα, IL-1β and IL-6 produced in the affected tissue stimulate recruitment of leukocytes such as monocytes, lymphocytes and neutrophils from the blood. As mentioned above, most DCs do not circulate through the blood and instead are inherent in peripheral tissues, especially in the skin and mucosal surfaces, where threats from pathogens and other environmental agents are greatest. Upon maturation, DCs change from phagocytes to professional antigen-presenting cells, with an altered chemokine receptor repertoire that promotes increased cell motility and tropism for lymph chemokines [177]. All DC subsets and other antigen-presenting cells such as macrophages, express an array of PRRs, which detect molecular patterns of invading microorganisms or endogenous “danger” signals such as those from damaged cells. Intracellular and cell surface PRRs sense a wide range of molecules, including proteins, carbohydrates, lipids and nucleic acids. Of the many pro-inflammatory stimuli that can induce maturation, most appear to act through TNFα and IL-1β [178,179,180]. Maturation results in several phenotypic changes to enhance uptake of antigen and subsequent presentation in draining lymph nodes, including upregulation of MHC class II and co-stimulatory molecules CD80, CD83 and CD86. Additionally, mature DCs downregulate expression of chemokine receptors CCR1, CCR2, CCR5 and CXCR1, required for pro-inflammatory chemotaxis in the tissues, while exhibiting enhanced expression of CCR7 [177].

Migratory leukocytes, however, are not the only cells undergoing phenotypic changes upon inflammatory stimuli. The lymphatic endothelium also adopts a radically different transcriptional profile, to selectively enhance lymphatic trafficking of specific immune cells [96,97]. ICAM-1 and VCAM-1 expression increases dramatically following TNFα stimulation, and antibody blockade or genetic deletion of these cell adhesion molecules or their integrin ligands LFA-1 and VLA4 significantly reduces exit of DCs from the skin and lymphatic migration to draining lymph nodes [96,100,101], thus impairing their capacity to induce CD8^+^ T-cell priming [98]. ICAM-1 accumulates on microvilli projections surrounding adherent DCs in a manner dependent upon the conformational change of β2 integrin on the DC, promoting DC crawling over the lymphatic endothelial basolateral surface and transendothelial migration [102]. Formation of such endothelial transmigratory cups is induced through ligation of the lymphatic endothelial receptor LYVE-1 engaging with its ligand hyaluronan (HA) [91], organized in a dense 400–500 nm thick glycocalyx on the DC surface by the leukocyte receptor CD44 [92]. Disruption of LYVE-1:HA interactions by gene deletion, antibody blockade or depletion of the HA glycocalyx impair DC adhesion, transmigration and lymphatic trafficking, resulting in diminished antigen-specific T-cell immune responses in draining lymph nodes [91]. Additionally, binding of HA to LYVE-1 induces disruption of the VE-cadherin-lined endothelial cell junctions, thus further aiding diapedesis [93]. Given that CD44-anchored HA extends beyond the sphere of interaction of smaller surface molecules such as β2 integrin (with an extracellular domain of 20 mm, [181,182]) or even selectins (50–100 nm, [183,184]), it is tempting to speculate that LYVE-1:HA interactions form the first adhesive contacts between migratory DCs and lymphatic endothelium. Additionally, as has been shown with tumour cell glycocalyces [185], the extended dimensions of CD44:HA:LYVE-1 complexes might constrain the lateral diffusion of smaller molecules, corralling them to promote clustering and subsequent integrin-mediated adhesion.

At present, it is unknown whether lymph-migrating T-cells also employ integrins when exiting inflamed peripheral tissue or indeed possess a HA-rich glycocalyx. In addition to the molecules facilitating homeostatic trafficking, such as MR [86,87,88] and CLEVER-1 [89,90], it is likely that specific leukocyte subsets are recruited by a number of additional receptors, including ALCAM (CD166) [104,105], L1CAM (CD171) [106], CD99 and CD31 [107]. E-selectin (CD62E) and P-selectin (CD62P) are also upregulated in inflamed lymphatic endothelium, albeit transiently in the case of the former [96,97], and the functional significance of this has yet to be demonstrated.

#### 3.1.2. Chemokines

In addition to adhesion molecules, increased immune cell recruitment to lymph nodes via afferent lymphatic vessels is further supported by chemokines. CCL21 is not only a homeostatic chemokine but is also upregulated during acute inflammation in both mouse and human lymphatic endothelia [75,186,187,188]. Enhanced secretion of CCL21 from LECs triggers integrin activation on DCs, stimulating DC translymphatic migration by chemotaxis [186].

As well as promoting inflammation-induced lymphatic trafficking of DCs, CCL21 and CCR7 also support efficient egress of CCR7^+^ T-cells from acutely inflamed peripheral tissue. CCR7 deficiency in splenocytes adoptively transferred into inflamed skin (following application of Complete Freund’s Adjuvant, CFA) resulted in reduced trafficking of CD4^+^ T-cells and B-cells by around 80% and 70% respectively [69]. Additionally, in a Th1-mediated antigen-specific delayed type hypersensitivity (DTH) model in mice, CCR7 deficiency was found to cause an accumulation of regulatory T-cells (T-regs) in the inflamed skin, which was accompanied by enhanced suppression of inflammation [70]. As the ratio between effector T-cells and T-regs at inflamed sites is a critical determinant for the outcome of the inflammatory response [189] egress of T-cell subsets must be carefully regulated, and, given that some T-cells are permitted to migrate in a CCR7-independent manner, other chemokines are clearly involved in such extra levels of selectivity.

As under steady-state conditions, S1P plays a crucial role in mediating retention of T-cells in inflamed sites, and tissue concentrations of this lipid mediator increase after initiation of an inflammatory response induced by alloantigen or viral antigen, coupled with a decrease in concentration of its precursor, ceramide [83]. This is coincidental with an increase in the early leukocyte activation marker and C-type lectin, CD69, on virus-specific CD8^+^ T-cells [114]. CD69 directly interacts with S1P_1_, resulting in mutual inhibition of surface expression and S1P_1_ degradation, coupled with transcriptional downregulation. Thus, such T-cells are retained in the skin and long-lived adaptive immune memory may be generated in peripheral tissue, to guard against future invasion by the same pathogen.

Lymphatic endothelium responds to inflammatory stimuli by increasing synthesis and secretion of a variety of chemotactic factors in addition to CCL21 and S1P. These include CCL1, CCL2, CCL5, CCL20, CXCL12 and CX3CL1 (chemotactic for DCs, monocytes and T-cells bearing the receptors CCR8, CCR2, CCR5, CXCR4 and CX3CR1 respectively) and CXCL1, CXCL2, CXCL5 and CXCL8 (neutrophil chemokines, acting through CXCR1 and CXCR2) [96,97,99]. Expression of CXCL12 in resting lymphatic endothelium is sparse and the CXCL12-CXCR4 chemokine-receptor pair have been shown to be redundant in steady-state T-cell trafficking [83]. In contrast, expression of CXCL12 mRNA has been shown to be induced in response to TNFα in cultured human dermal LECs [107], as well as at the protein level on lymphatic vessels within mouse dermis following hapten application [108]. Furthermore, CXCR4 is highly expressed in cutaneous MHC class II^+^ DCs, and a pharmacological antagonist (4-F-benzoyl-TN14003) to CXCR4 impairs lymph node migration of both dermal DCs and Langerhans cells by up to 50% during the sensitization phase of contact hypersensitivity. Like CCL21, CXCL12 binds heparan sulphate through a basic region at the N-terminus and hence may establish haptotactic gradients in tissues [109,110]. Lymphatic-derived CXCL12 may also serve to enhance DC survival, as CXCR4 activation has been shown to increase DC viability [111].

Like CXCL12, CX3CL1 is largely absent from resting lymphatic endothelium but is dramatically induced following stimulation with pro-inflammatory cytokines, particularly TNFα, through de novo RNA and protein synthesis in both primary LECs and in intact vessels within mouse and human dermis [96,115]. CX3CL1 is an unusual chemokine in that it is synthesized as a large (373 amino acids) type I integral membrane protein, comprising an extracellular domain that contains a novel C-X-X-X-C chemokine motif and an extended mucin-like stalk [190]. Expressed on several cell types, including activated vascular endothelium, CX3CL1 can exist in either membrane-anchored or soluble forms [190,191]. The membrane-bound form of CX3CL1 supports extravasation from the blood through inducing shear-resistant adhesion of leukocytes to blood endothelium, in a manner independent of activation of either integrins or Gαi proteins [190,192,193,194,195]. Additionally, soluble forms of CX3CL1 (generated by proteolytic cleavage with the disintegrin and metalloproteinases ADAM10 and ADAM17) promote conventional integrin-mediated chemotaxis [196,197]. The sole receptor, CX3CR1, is widely expressed by leukocytes, including CD14^+^ cells of the monocyte/macrophage/DC lineage and subsets of tissue-resident DCs and epidermal Langerhans cells [194,198,199]. In inflamed lymphatic endothelium, CX3CL1 is rapidly shed from the cell surface in a predominantly basolateral direction, promoting transmigration of monocyte-derived DCs across monolayers of primary LECs in vitro and supporting trafficking to draining lymph nodes in vivo [115]. However, dual blockade of CX3CL1 and CCL21 does not yield an additive effect in transmigration assays, and targeting CX3CL1:CX3CR1 interactions by neutralizing antibodies does not result in a “log-jam” of DCs around lymphatic capillaries, as has been reported following disruption of CCL21:CCR7 [63,64] or ICAM:LFA1 interactions [96]. This suggests that CCL21 and CX3CL1 act sequentially, with CX3CL1 providing guidance for DC migration through the interstitium by fluid phase gradients, while CCL21 forms haptotactic gradients and facilitates docking to lymphatic endothelium.

LECs release basolateral CD9^+^ CD63^+^ exosome-rich endothelial vesicles, both constitutively and also in greater numbers in response to pro-inflammatory cytokines, leading to the formation of peri-lymphatic halos around initial vessels [116]. Proteomic analysis of these exosomes has revealed that the cargo proteins are predominantly associated with a motility-promoting function, such as chemokines, actin cytoskeleton regulatory proteins, motor proteins and adhesion molecules, and include membrane-anchored CX3CL1. These exosomes enhance formation of cellular protrusions in human monocyte-derived DCs and promote their transit across lymphatic endothelium, in a CX3CL1-dependent manner. Such exosome-driven transmigration is in co-operation with CCL21, and, as CCL2 and CCL5 are also enriched within these lymphatic-derived exosomes, it is likely that they selectively aid lymphatic homing of multiple leukocyte subsets [116].

As blood and lymphatic capillaries are so closely apposed within peripheral tissues, it is possible that lymphatic endothelial-derived chemokines act in a paracrine fashion to recruit more monocytes and T-cells from the blood after transfer to the luminal face of blood vascular endothelium [200]. However, only CCL21 and CX3CL1 have been found to be secreted basolaterally, while CCL2, CCL5 and CCL20 are secreted predominantly from the luminal surface, into the afferent lymph [74,115,186]. Furthermore, local accumulation of such inflammatory chemokines has been shown to be detrimental to efficient leukocyte entry to lymphatics, with the atypical chemokine receptor ACKR2 playing a crucial role in preventing such accrual (reviewed by [201]). ACKR2 exhibits some sequence homology with other chemokine receptors and binds to at least 12 different CC chemokines. However, it internalizes bound ligand and targets it for intracellular degradation, rather than conventional signaling. ACKR2 is expressed in dermal LECs, as well as in other tissues which perform a barrier function [117], ensuring that chemokines including CCL2 and CCL5 are absent from the basolateral surface of lymphatic vessels and thus preventing congestion around sites of entry [118]. Furthermore, ACKR2 is upregulated under inflammatory conditions, permitting CCR2^−^ CCR5^−^ CCR7^+^ mature DCs (in preference to CCR2^+^ CCR5^+^ CCR7^−^ immature DCs) to adhere to lymphatic endothelium [119,177]. Moreover, studies using ACKR2-deficient mice have demonstrated the importance of this molecule in allowing effective resolution of cutaneous inflammatory responses [120]. Efficient migration of DCs and Langerhans cells from skin is also aided by an additional atypical chemokine receptor, ACKR4, expressed by keratinocytes and a subset of LECs [122]. This receptor scavenges stromal-derived CCL19, preventing de-sensitization of CCR7 and preserving the responsiveness of DCs and Langerhans cells to CCL21.

#### 3.1.3. Neutrophils

Neutrophils are the first leukocyte population to be recruited from the blood into inflamed and/or injured peripheral tissue, in response to cytokines, such as TNFα and IL-1β, and chemokines such as CXCL8 (reviewed in [202]). They exhibit highly effective anti-microbial activity, through a combination of phagocytosis and release of cytotoxic granules and extracellular traps, before undergoing apoptosis within the tissue. Dead neutrophils are subsequently cleared by macrophages, via the lymphatics. However, it is now accepted that neutrophils are not always the short-lived infantrymen they were originally believed to be, and lymph-migrating neutrophils (albeit the minority) have an extended lifespan beyond the usual T_1/2_ ~ 6–12 h. In mice, following dermal immunization with either Mycobacterium bovis bacillus Calmette-Guerin (BCG) or peptide antigen (ovalbumin), neutrophils are the rapid responders responsible for capturing bacilli or antigen and transporting them to draining lymph nodes via afferent lymphatics [203,204]. Ovalbumin-pulsed neutrophils display a DC-like phenotype, presenting antigens in an MHC class II-dependent manner to induce proliferation of antigen-specific T-cells and regulate adaptive immune responses [205].

Clearly, however, neutrophil entry into the lymphatics must be carefully regulated and balanced, tailored to individual types of infection and inflammation. In a study of skin inflammation elicited by topical application of CFA, lymph migratory neutrophils were found to be exclusively CCR7^+^, with CCR7^−/−^ mice exhibiting around 60% fewer neutrophils in draining lymph nodes [121]. Moreover, entry of neutrophils to TNFα-stimulated cremaster lymphatic vessels was almost completely abolished in CCR7^−/−^ mice [206]. In contrast, following intradermal administration of killed *Staphylococcus aureus*, lymph-borne neutrophils did not require CCR7 for skin egress but rather CXCR4 and CD11b, (αM integrin subunit, associating with β2), [207]. Curiously, pertussis toxin did not inhibit neutrophil migration to lymph nodes [207], suggesting that CXCR4 acts in a Gαi-independent manner [112,113]. Additionally, ICAM-1 blockade did not reduce neutrophil egress from skin in response to *S. aureus*, and thus CD11b may regulate lymphatic migration through other interactions such as JAM-C, as has been shown in neutrophil extravasation [208,209].

β2 integrin was found to be similarly important in mediating lymphatic migration of neutrophils following infection with *M. bovis* BCG, through both ICAM-1-dependent and -independent mechanisms [99]. In vitro studies using primary human dermal LECs and peripheral blood-derived neutrophils indicate involvement of additional adhesion and junctional molecules in neutrophil adhesion and diapedesis, specifically E-selectin, CD99 and VCAM-1 [99].

Adhesion triggers local release of neutrophil elastase, matrix metalloproteinases MMP8 and MMP9, and the arachidonate-derived chemorepellent lipid 12-hydroxyeicosatetraenoate (12(S)HETE), which induce extremely targeted endothelial junctional retraction [99,210]. These act as transient portals, permitting ~ 10-fold more rapid migration of neutrophils than DCs before resolving spontaneously, with no permanent damage to the integrity of the lymphatic endothelium. It is interesting to note that although TNFα-stimulated human LECs upregulate expression of various neutrophil chemokines, namely CXCL2, CXCL5 and CXCL8 [96,97], these are secreted almost exclusively from the luminal surface, with only CXCL8 directing transmigration [99].

#### 3.1.4. Immunomodulation by Peripheral LECs

Adhesion interactions between DCs and LECs do not only serve to facilitate trafficking but may also modulate DC function and subsequent immune responses. In vitro studies have revealed that αMβ2 integrin:ICAM-1-mediated interactions between human immature monocyte-derived DCs and TNFα-stimulated LECs resulted in a reduction in surface levels of the co-stimulatory molecule CD86 on these DCs [211]. Subsequently, such “LEC-educated” immature DCs exhibited an impaired ability to activate T-cells in a mixed lymphocyte reaction. Crucially, this blunting of DC-mediated responses only occurred in the absence of lipopolysaccharide (LPS), an example of a pathogen-derived signal, suggestive of a role for LECs in preventing undesired immune reactions when there is no threat of infection.

The findings from a later investigation in transgenic mice further support the concept of immune regulation by LECs, where DC maturation was suppressed by the anti-inflammatory effects of prostaglandin, secreted by LECs in inflamed skin [212]. The transgenic mice used in this study exhibited a VEGF-C-induced expansion of lymphatic vessels within the skin, which, in addition to the preponderance of immature, tolerogenic DCs, established an immune-inhibitory microenvironment characterized by CD8^+^ T-cells with decreased effector function, increased numbers of T-regs, reduced levels of inflammatory cytokines including TNFα, IL-6 and IFN-γ, and increased secretion of the anti-inflammatory cytokine TGF-β1. Such a response could be expected to be most physiologically relevant during resolution, following inflammation-induced expansion of the lymphatic network and acting to limit chronic inflammation [213].

### 3.2. Intraluminal Crawling in Initial Lymphatic Capillaries

Upon entering afferent lymphatic vessels, leukocytes actively crawl in a semi-directed manner along the luminal surface of initial capillaries, where the lymph flow alone (0–30 µm/s) is insufficient to propel them towards the larger collecting vessels and draining lymph node [214,215,216,217]. Surprisingly, DCs crawl at an even slower rate than average lymph flow in these small vessels, and downstream-directed DC migration is lymph flow independent [82]. Instead, DC crawling is directed by CCL21 and CCR7, with intralymphatic CCL21 gradients induced by the low lymph flow. It is possible that CCL2:CCR2 may play a role at this stage too, as “cords” of DCs were visible inside lymphatic vessels of CCR2-deficient mice, suggestive of a defect in intraluminal crawling rather than due to impaired vessel entry [123]. Such lymphatic cords are also apparent when increased levels of endogenous HA are incorporated into the CD44-held glycocalyx on DCs, by antibody-induced potentiation of CD44:HA interactions [92]. Adhesion of these DCs to lymphatic endothelium was found to be greatly enhanced; however, transit to draining lymph nodes was impaired and such cells remained stuck in initial capillaries. As the lymphatic HA receptor LYVE-1 is expressed on both the luminal and basolateral surfaces of these lymphatic vessels [218], this may be indicative of a role for the LYVE-1:HA:CD44 axis in mediating intraluminal crawling, especially as CD44 is distributed predominantly within the pro-adhesive uropod of crawling DCs [91,92].

In addition to chemotactic cues, DC crawling within lymphatic vessels is further supported by the Rho-associated protein kinase (ROCK), which promotes reorganization and contraction of actomyosin filaments, thus permitting cellular contraction and de-adhesion at the uropod [219]. ROCK plays a minor role in intralymphatic DC migration from resting tissues but is indispensable during tissue inflammation, mediating de-adhesion from ICAM-1 [124], which is expressed on both basolateral and luminal surfaces of inflamed lymphatic endothelium [96].

Like DCs, T-cell crawling within lymphatic vessels of inflamed mouse skin is dependent upon interactions with LFA-1 integrin interactions with ICAM-1 [103]. Intriguingly, intravital microscopy studies of inflamed mouse skin during hypersensitivity responses have also revealed that antigen-experienced effector T-cells and DCs frequently interact with one another within afferent lymphatic capillaries [220]. Such interactions between polyclonal T-cells and adoptively transferred DCs that were not hapten-experienced were found to be short-lived, with DCs interacting with multiple T-cells simultaneously, in an MHC class II-independent manner. However, contacts between antigen-specific T-cells and cognate antigen-bearing DCs required MHC class II and lasted much longer (>30 min in over 60% of DC:T-cell contacts), typically forming between a DC and a single T-cell. Thus, afferent lymphatic vessels may play a role in modulating and supporting adaptive immune responses during leukocyte transit, even before they reach the lymph node.

Once leukocytes are in the larger pre-collector and collector lymphatic vessels, lymph flow is much faster due to the presence of contracting smooth muscle cells surrounding the vessels and intraluminal valves that prevent backflow of lymph. Here, leukocytes move in a passive manner, at speeds of around 1.2 mm/min, until encountering a lymph node [78,82,103,214,215,216,217].

### 3.3. In the Inflamed Lymph Node

In lymph nodes draining the site of inflammation, leukocyte accumulation is dramatically enhanced, while lymphocyte exit is transiently blocked [221,222,223]. These inflammation-induced changes, as well as the highly specialized architecture of the lymph node, all increase the probability of antigen presentation to the cognate lymphocyte.

Lymph nodes undergo considerable expansion during acute inflammation, with endothelial and mesenchymal lymphoid stromal cells driving this extensive remodeling. Infection and inflammation induce rapid transcriptional responses in the fibroblasts, LECs and blood endothelial cells (BECs) of the lymph node, through expression of genes involved in cytokine signaling, chemoattraction, adhesion, antigen processing and presentation, as well as producing growth factors and matrix metalloproteinases (MMPs) regulating vasculature expansion and modifications to extracellular matrix [224]. Indeed, transcriptional analysis has revealed that all stromal cell subsets in draining lymph nodes respond to infection of the peripheral tissue with different kinetics for distinct pathogens [225]. Moreover, although the transcriptional program is transient, expanded FRC networks persist, to support successive immune responses.

#### 3.3.1. Entry via Afferent Lymphatics

Inflammation-induced rapid responses in the lymph node are mediated by soluble mediators secreted from peripheral tissue and travelling in afferent lymph, then rapidly transported from the SCS through fibroblastic reticular conduits [36]. Evidence of this “remote control” first came from studying *plt/plt* mice, which lack CCL19 and one of the two CCL21 isoforms in mouse, CCL21^Ser^. Investigators found that defective T-cell homing could be restored by cutaneous injection of CCL21 or CCL19 and within 2 h of injection, these exogenously applied chemokines were rapidly transported to HECs and presented on the apical surface following transcytosis [32,39]. Endogenously derived CCL2 secreted from undetermined cells within inflamed skin has also been shown to be presented on HEVs, mediating integrin-dependent arrest of rolling monocytes from the blood circulation and thus enhanced recruitment to the lymph node [223]. Primary dermal LECs secrete the monocytic and T-cell chemoattractants CCL2, CCL5, and CCL20, as well as the neutrophil-chemoattractants CXCL2, CXCL5 and CXCL8, predominantly from the luminal surfaces and thus it is highly likely that peripheral LECs contribute to elevated levels of these chemokines within lymph nodes draining inflamed tissue [96,99].

Chemokines CXCL9 and CXCL10 have also been found to mediate inflammation-induced recruitment of monocytes and IFN-γ-producing CD4^+^ T-cells respectively via HEVs, through engagement with CXCR3 [20,226]. However, it is unclear whether CXCL9 and CXCL10 are produced by HECs or other cells, either within the lymph node or upstream in peripheral tissue. It is likely that these are mainly derived from lymph node LECs, in particular the SCS floor LECs and the Ptx3-LECs of the medulla and paracortical sinus (Figure 3). Following oxazolone-induced skin inflammation, both subsets of LECs in draining lymph nodes exhibit a type I interferon response, with increased expression of *cxcl9*, *cxcl10* and *Irf7* transcripts, suggestive of recruitment of CXCR3^+^ DCs, NK cells, effector T-cells and monocytes [136]. SCS floor LECs also upregulate *Ccl20*, [130,136] whereas Ptx3-LEC exhibit enhanced expression of *Ccl2* and the broader chemoattractant *Ccl5*, implying that chemotactic regulation in the SCS and medulla diverge in inflammation.

Specific subsets of lymph node LEC may also regulate neutrophil homing. Single-cell analysis of human lymph nodes revealed that both SCS floor LECs and those in the medullary sinus express transcripts encoding the neutrophil chemoattractants *Cxcl1*–*Cxcl5* [130]. Furthermore, neutrophil adhesion to LECs in the medullary sinuses was shown to be supported by the C-type lectin CD209 (DC-SIGN), through interactions with Lewis^X^ carbohydrates, abundantly expressed on neutrophils. As discussed earlier, neutrophil migration to draining lymph nodes is a critical component of defence against infectious agents [227], and such recruitment and intra-nodal positioning of neutrophils could kill bacteria in the medulla, thus preventing further dissemination into the circulation. Additionally, skin-egressing neutrophils, recruited in response to *S*. aureus infection, are critical for augmenting lymphocyte proliferation in draining lymph nodes [207]. In another study, following subcutaneous injection of ovalbumin and CFA, migratory neutrophils were found to secrete T_H_1 cytokines, including TNFα, once in draining lymph nodes, [204]. However, the oxidative burst that is essential for killing bacteria is inhibited in neutrophils that have phagocytosed BCG and thus they are either protecting live bacilli, delaying or impairing the induction of the adaptive immune response, or they are polarizing the T-cells response toward a T_H_2 profile, to limit a potentially harmful proinflammatory cytokine storm [203]. Therefore, it is difficult to determine whether neutrophils are acting in the host’s best interest or are simply being hijacked by the pathogen, and this likely depends upon the individual pathogen, as well as the host’s immune system.

Ptx3-LECs of the medullary sinus have been shown to express numerous genes suggestive of responsiveness to the lymphangiogenic factor VEGF-C, including *Vegfr3* (*flt4*) and its co-receptor *Neuropilin 2* (*Nrp2*), [136]. VEGF-C is a well-known instigator of lymphangiogenesis, during development and in the adult [228,229,230,231], with expansion of LYVE-1^+^ medullary and paracortical sinuses a feature of the acute inflammatory response to topical application or immunization with antigens [232]. Indeed, such lymphangiogenesis results in HEVs and lymphatic vessels being directly apposed, permitting cross-talk between the two, through B-cells and lymphotoxin-β receptor (LTβR). Such remodeling of the lymphatic network within lymph nodes serves to support the increase in DC migration from inflamed peripheral tissue [233].

Single-cell transcriptome analyses have revealed that although lymph node LEC subsets are conserved between mouse and human, additional subsets of SCS ceiling LECs are apparent in the latter. The reasons for this greater heterogeneity and degree of specialization may be due to humans experiencing more infections than mice which are housed under pathogen-free conditions [130,136]. Additionally, human lymph nodes exhibit a more complex architecture than those in mice, with invaginations of the capsule termed trabecular sinuses [141]. In peripheral tissue, lymph fluid shear stress contributes to endothelial junction stability and lymphatic endothelial integrity [234] and thus, turbulent flow and higher shear stress could well be anticipated to affect the transcriptional profile of trabecular sinus LECs in lymph nodes as well.

#### 3.3.2. Entry via HEVs

As mentioned above, immune cell entry to the lymph node from the blood is predominantly dictated by events in the periphery. Inflammatory chemokines such as CCL3, CXCL8 and CXCL9 are displayed on the apical surface of HECs, to support recruitment of monocytes, in addition to constitutively recirculating lymphocytes [36]. This route is also taken by plasmacytoid DCs (pDCs) and precursors of conventional DC [235].

pDCs (reviewed in [236]) are much more abundant in the blood than conventional DCs and preferentially migrate to secondary lymphoid organs including lymph nodes, entering via HEVs, rather than to inflamed peripheral tissue [237]. Human pDCs are rarely detected in heathy lymph nodes and constitutively express only very low levels of CCR7, albeit sufficient to support migration towards CCL19 and CCL21 in vitro. Recruitment is markedly increased during inflammation, with pDCs detected around and within HEVs of mycobacteria-infected lymph nodes [235]. As with T-cells, rolling and firm adhesion of pDCs to HECs is mediated by L-selectin and integrins, respectively, with CCR7-dependent signalling. Lymph node homing of adoptively transferred CCR7-deficient pDCs in mice is impaired, both under steady-state conditions and following stimulation with the TLR7/8 agonist, R848, which induces a dramatic increase in CCR7 expression [29]. However, other chemotactic guidance cues may be involved too, such as CXCR3, CXCR4, CCR5, CCR10 and the chemerin receptor, ChemR23 [238,239,240,241], although there may well be differences between mice and humans as to which of these are most critical. Human pDCs express CXCR3 [235] and secretion of the ligands of this chemokine receptor, CXCL9 and CXCL10, is induced in human umbilical vein endothelial cells (HUVECs) in response to stimulation with TNFα or IFNγ [242]. Furthermore, CXCL9 has been detected on HEVs of inflamed lymph nodes in mice, mediating extravasation of monocytes [226]. In addition to CCR7, in mice (but not humans), CCL21 can bind to CXCR3 and induce signaling [243,244]. Conversely, pDCs in humans but not mice express ChemR23, with chemerin expressed on the luminal surface of HECs, contributing to lymph node homing [241].

The main characteristic of pDCs is the ability to produce large amounts of Type I IFNs, which can promote T-cell survival and protect antigen-activated cytotoxic CD8^+^ T-cells from antigen-induced cell death [235,245,246]. pDCs also secrete CCL3, a chemoattractant that recruits effector T-cells, particularly Th1-type, further promoting an inflammatory environment within draining lymph nodes [240].

### 3.4. Exiting Inflamed Lymph Nodes via Efferent Lymphatic Vessels

#### 3.4.1. Initial Retention

Studies in sheep have shown that subcutaneous injection of antigen leads to an acute but transient reduction in the output of lymphocytes in efferent lymph of the draining popliteal lymph nodes [247]. Multiple cortical sinuses are situated adjacent to HEVs and under steady-state conditions, some lymphocytes access these sinuses within minutes of entering the lymph node and exit swiftly [140,144]. However, upon entry to an inflamed lymph node and in response to inflammatory mediators such as Type I interferons, lymphocytes rapidly upregulate CD69 and are prevented from accessing cortical sinuses [140,248]. CD69 also interacts with S1P_1_, inducing a receptor conformation that is similar to the ligand-bound state, promoting internalization and degradation, [248,249]. This is coupled with transcriptional down-regulation of S1P_1_ which occurs in response to T-cell receptor signaling during early activation [171]. These responses, in addition to decreased surface expression of CCR7 [172], all promote retention of the newly activated T-cell. Such modulation by CD69 and S1P_1_ is similar to that occurring in peripheral tissues (discussed above) to retain tissue-resident memory T-cells for future adaptive immune responses [114].

Circadian rhythm affects various aspects of the immune system, including lymphocyte recirculation (reviewed by [250,251]). In mice, neural inputs to β_2_-adrenergic receptors expressed on lymphocytes were found to reduce the frequency of lymphocyte egress at night, and the resulting accumulation of lymphocytes within lymph nodes led to enhanced antibody responses to intradermally administered hapten antigen [252]. Moreover, these lymphocyte-expressed β_2_-adrenergic receptors can directly interact with CCR7 and CXCR4, again to promote retention [253]. Thus, physiological inputs from adrenergic nerves can contribute to immune regulation by the nervous system and may aid the fine-tuning of lymphocyte retention-versus-egression in lymphoid tissue. Of note, lymphocyte distribution between lymphoid tissue and blood exhibits an opposing diurnal fluctuation in humans to those in mice, reflecting the differences in diurnal/nocturnal behaviours of the two species [254].

#### 3.4.2. Egression

The initial decrease in lymphocyte output is followed by a phase of enhanced egress. CD4^+^ T-cells usually exit in advance of CD8^+^ T-cells, although the ratio and timing appear to depend largely upon the nature of antigen or infectious agent [9,10,52,221,255,256,257,258,259,260,261,262,263]. Activated T-cells only transiently express CD69 and after several rounds of division display a marked reduction in expression [264], with a reciprocal increase in surface levels of S1P_1_ [171,172]. Additionally, expression of tenascin-C is upregulated and, through binding α9 integrin on LECs of the medullary and cortical sinuses, induces secretion of S1P to further promote lymphocyte egress [265]. Indeed, antibody blockade of α9 integrin-mediated signaling was found to reduce lymphocyte egress in murine experimental autoimmune encephalomyelitis, resulting in improved clinical scores and attenuated pathology [265]. It is likely that further adhesion molecules are involved in this balancing act of modulating T-cell transit time, for example ICAM-1 and its integrin ligands αLβ2 [266], (Table 1). However, other molecular regulators in this process remain to be identified.

DCs are absent from efferent lymph. The lymph node is the “end of the line” for them, and once they have presented antigen to a cognate T-cell, they die, mostly likely through FAS/FAS-ligand-mediated apoptosis [267,268]. Indeed, photoconversion experiments to track migratory dermal DCs following either chemical stress (application of 1:1 acetone:dibutyl phthalate) or mechanical injury (adhesive tape stripping) revealed that in the lymph node, skin-derived DCs expressed the highest levels of the apoptosis marker caspase-3 [269]. In contrast, virtually no neighbouring T- or B-cells displayed caspase activity. Such “shooting the messenger” frees the activated T-cell to proliferate but also provides a mechanism for terminating the response to a particular antigen, which would be damaging if allowed to continue over an extended period of time. Additionally, DCs expressing peptides derived from endogenous pathogens such as viruses may be eliminated by cytotoxic T-cell-induced apoptosis [270].

## 4. Chronic Inflammation

Unresolved acute tissue inflammation can lead to a persistent inflammatory state in which tissue damage and fibrosis occur. Such chronic inflammation contributes to numerous diseases, including arthritis, asthma, atherosclerosis, autoimmune disorders, diabetes, cancer, and those associated with ageing [271]. IL-6 contributes to the acute phase of inflammation but if levels remain high, this cytokine promotes survival and growth of lymphocytes and macrophages. Thus, whereas acute inflammation is characterized by a predominantly polymorphonuclear leukocyte infiltrate, chronically inflamed tissue contains mainly mononuclear immune cells.

Under conditions of prolonged inflammation, tissue fibroblasts drive the recruitment and formation of ectopic clusters of lymphocytes and myeloid cells, termed tertiary lymphoid structures (TLSs) because of their resemblance to secondary lymphoid organs (reviewed in [272,273]). Unlike lymph nodes, these structures are not encapsulated and possess variable degrees of organization, from simple B- and T-cell aggregates to more complex structures complete with HEVs, B-cell follicles with follicular DCs, germinal centres and occasionally LECs. The function of these TLSs is likely as an early, local defence against microbes, and their constituent cells dissipate once the pathogen is eliminated. However, the high number of adaptive immune cells in these structures can exacerbate autoimmune diseases, with TLSs a feature in numerous disorders including Hashimoto thyroiditis, diabetes and multiple sclerosis.

### 4.1. Composition of Afferent Lymph Draining Chronically Inflamed Tissue

Th1 and Th17 T-cell subsets, secreting IFN-γ and IL-17 respectively, are responsible for driving the development and severity of prolonged tissue inflammation and these effector/memory cells also constitute the majority of leukocytes in afferent lymph [69]. CFA-induced cutaneous inflammation provokes development of granulomas in sheep (after 3–4 weeks) and mice (2–3 weeks), with enhanced lymphocyte exit in comparison to that measured during acute inflammation. Indeed, such egress of inflammatory T-cells from chronically inflamed tissue is likely to be an important control point of the inflammatory response. However, remarkably little is known about the molecules controlling this clearance. In contrast to trafficking during acute inflammation, entry of both CD4^+^ and CD8^+^ T-cells to afferent lymphatic vessels draining chronically inflamed skin occurs in a manner largely independent of both CCR7 [69] and CXCR4 [274]. Nevertheless, such egress is sensitive to pertussis toxin treatment, demonstrating that this is not a passive process and Gαi-coupled chemoattractant receptors are involved, with some contribution from S1P receptors and other as yet unidentified Gαi-coupled receptors [69].

### 4.2. Lymphatic Remodelling in Chronically Inflamed Tissue

Persistent inflammation of the airways in mice infected with *Mycoplasma pulmonis* leads to widespread remodeling of both blood and lymphatic vessels, including enlargement of capillaries into venules, and lymphangiogenesis [275]. TNFα has been shown to drive the former of these processes directly; however, the effects on lymphangiogenesis likely require other inflammatory mediators secreted from leukocytes. Inflammation also induces changes in lymphatic endothelial cell junctions. As described above, the endothelial cells of initial lymphatic capillaries in uninflamed tissue have discontinuous button junctions, in contrast to the conventional zipper junctions between endothelial cells in blood capillaries and collecting lymphatic vessels [60]. In contrast, 14–28 days after inflammation-induced *M. pulmonis* infection, zipper junctions were found to replace the button junctions. Moreover, such changes in junction arrangement could be reversed by treatment with the anti-inflammatory steroid, dexamethasone. These junctional alterations in inflamed lymphatic vessels may well be responsible for impaired fluid entry, edema and poorer clearance of inflammatory leukocytes. However, as these junctions exhibit such plasticity, this represents an attractive target for alleviating chronic inflammation.

## 5. Resolution

Once the proinflammatory response to injury or infection has begun to subside, inflammatory leukocytes switch to a reparative phenotype and cell debris is removed via afferent lymphatics from the inflamed tissue, in an attempt to restore normal tissue architecture and function. Such clearance is aided by inflammatory-associated lymphangiogenesis and lymphatic remodeling within the affected peripheral tissue (reviewed by [276]). In mouse skin, a dramatic expansion of lymphatic vessels is apparent following inflammation induced by bacterial agents (LPS, lipoteichoic acid and muramyl dipeptide), [277]. This is concomitant with increased macrophage infiltration and expression of macrophage-derived lymphangiogenic growth factors VEGF-C, VEGF-D and VEGF-A. Moreover, antigen clearance and resolution of inflammation were found to be dependent upon this reparative macrophage-driven lymphangiogenesis.

VEGF-C-driven lymphangiogenesis also improves lymphatic clearance of neutrophils, macrophages and DCs in a mouse model of myocardial infarction, whereby the coronary artery of a mouse is ligated [94]. In this model, after the initial inflammatory phase of leukocyte recruitment from the blood, reparative macrophages and other leukocytes are cleared from the cardiac muscle. The mice recover, with limited fibrotic scarring, within 21 days. Critical to this resolution was expression of the HA receptor, LYVE-1 on lymphatic endothelium. Accumulations of CD68^+^ macrophages, as well as neutrophils and DCs were apparent in the cardiac muscles of LYVE-1-deficient mice, with a simultaneous reduction in numbers of these cells recovered from mediastinal draining lymph nodes. Moreover, at 21 days post-infarction, the hearts of LYVE-1^−/−^ mice displayed increased fibrosis and reduced cardiac output. This and other studies have demonstrated that LYVE-1 is an important mediator of lymphatic trafficking, through interactions with its ligand HA, expressed as a glycocalyx on DCs [91] and monocyte-derived macrophages [95]. However, as neutrophils do not synthesize an HA glycocalyx [207], it is unclear how LYVE-1-deficiency impedes trafficking of this lymph-migrating population. It may be that the neutrophils detected in the myocardial tissue were dead or dying and were awaiting clearance by macrophages.

Another molecule that regulates skin inflammation, resolution and lymphatic drainage is IL-7. This interleukin is predominantly expressed by lymphocytes but also by LECs from afferent capillaries [278]. Additionally, peripheral LECs express the two IL-7 receptor chains: IL-7Rα and the common cytokine receptor γ-chain (CD132), with endothelial-specific deletion of IL-7Rα found to result in edema and impaired lymphatic drainage in psoriasis-like skin inflammation in mice [279]. However, although systemic therapy with IL-7-Fc fusion protein improved lymphatic drainage from psoriatic skin, such treatment exacerbated edema and immune cell infiltration. In contrast, administration of an IL-7Rα blocking antibody did not alter lymphatic drainage but ameliorated edema and decreased immune cell infiltration in the psoriatic lesions. Hence, IL-7Rα signaling exhibits opposing roles in endothelial cells and lymphocytes, whereby its pro-inflammatory effects on immune cells override its anti-inflammatory effects that enhance lymphatic drainage.

Enhanced drainage of excess tissue fluid may be further supported through the remodeling of the lymphatic vessel glycocalyx. The composition of this glycan coating has not been so extensively studied as that of the blood vascular endothelium but is known to contain heparan sulphate, α-D-galactosyl moieties, α2,3-linked sialic acids and *N*-acetylglucosamine moieties [280]. Within 16 h of inflammatory stimulus to mouse cremaster muscle, this lymphatic endothelial glycocalyx is rapidly remodeled, with a marked reduction in heparan sulphate and α-D-galactosyl moieties. Interestingly, treatment with a heparanase inhibitor prevented glycocalyx degradation and led to reduced capacity of these lymphatic vessels to drain interstitial fluid, thus highlighting a heparanase-dependent mechanism for alleviating inflammation-induced edema.

Clearly inflammation-associated lymphangiogenesis and lymphatic remodeling provide promising therapeutic targets. An increased lymphatic network improves the efficiency of immune cell delivery to draining lymph nodes, as well as clearing dead cells and excess interstitial fluid from injured and edematous tissue. However, too much trafficking could overwhelm the immune surveillance machinery within the lymphatic nodes and thus such pro-lymphangiogenic signaling pathways must be carefully regulated [276].

## 6. Conclusions

Seventy years have passed since the first studies on leukocyte trafficking in the lymphatics, and our knowledge of the roles that the lymphatic system plays in immunological processes has expanded greatly. Improved methods in gene manipulation, genetic sequencing, super-resolution and intravital imaging techniques, monoclonal antibodies and flow cytometry to identify leukocyte subsets, as well as greater computing power and new animal models have all helped to accelerate this learning. We can now appreciate the complexity of lymph nodes, in which stromal cells including LECs provide directional cues to ensure that leukocyte subsets home to specific locations, sampling antigen and responding to inflammatory stimuli and potential pathogens while maintaining tolerance for non-harmful exogenous and endogenous antigens. Additionally, the multiple chemotactic and adhesion receptors and ligands that regulate migration of specific leukocytes are better understood. However, there still remain a great number of unanswered questions, especially those pertinent to how we can harness the power of the lymphatic system to alleviate the symptoms of excessive inflammation, such as edema, fibrosis and neoplasms. The afferent lymphatics remain an attractive target for boosting immune responses, for example during vaccine delivery and for drainage of excess fluid. Conversely, the lymphatic system is frequently subverted under pathological conditions, including cancer metastasis and in autoimmune disorders. Identification of new molecules and better characterization of the ones we already know about may allow us to develop more specific therapies in the future, with fewer off-target effects.

## Figures and Tables

**Figure 1 ijms-22-04458-f001:**
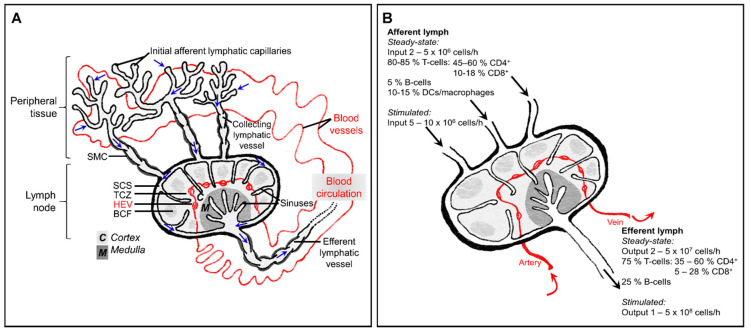
The interplay between the blood circulation and lymphatic system, and immune cell composition of afferent and efferent lymph. (**A**) In peripheral tissue, fluid and macromolecules that have leaked from the blood, as well as leukocytes, leave the interstitium and enter initial lymphatic capillaries. This lymph fluid (movement of which is indicated by blue arrows) then passes into larger collecting vessels which contain intraluminal valves and are invested with contracting smooth muscle cells (SMC), finally entering the lymph node at the subcapsular sinus (SCS). Fluid and low molecular weight particles (<70 kDa) percolate through the fibroblastic reticular network of the lymph node. Leukocytes cross the lymphatic endothelium into the lymph node parenchyma at subcapsular or medullary sinuses, or exit directly through the efferent lymphatic vessel. Lymphocytes enter the lymph node from the blood by way of high endothelial venules (HEV), then are guided by chemokines to areas of the lymph node, including the T-cell zone (TCZ) and B-cell follicle (BCF). Lymphocytes return to the blood circulation via the efferent lymphatic vessel and thoracic duct. (**B**) Distribution of leukocyte subsets in afferent and efferent lymph, from studies of ovine skin-draining lymph nodes. The cellularity of afferent lymph is much lower (5–10%) than that of efferent lymph under homeostatic conditions. Additionally, afferent lymph contains a more heterogeneous assortment of leukocytes, whereas only lymphocytes are found in efferent lymph.

**Figure 2 ijms-22-04458-f002:**
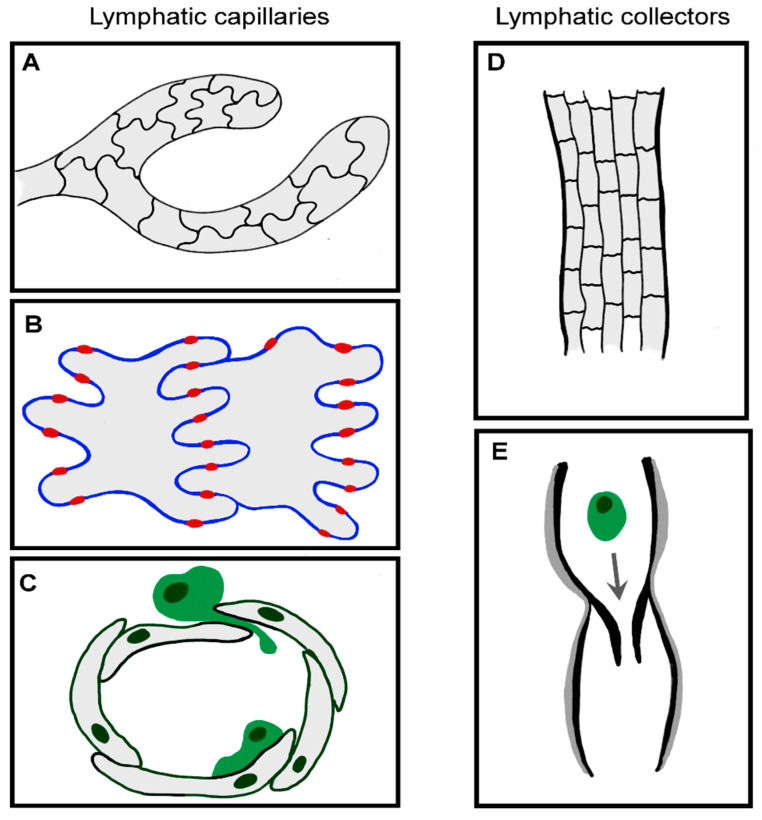
Functional specialization of LEC junctions. (**A**–**C**) Blind-ended capillaries of the initial lymphatics contain interdigitating oakleaf-shaped endothelial cells (**A**) held together with discontinuous button junctions, where adherens and tight junctional proteins provide the points of anchorage (red) in between flaps (**B**) through which fluid, macromolecules and migratory leukocytes enter without compromising the integrity of the endothelium (**C**) [60]. Once inside the vessel, leukocytes actively crawl along the luminal surface of endothelial cells, passing into pre-collector and collector vessels. (**D**,**E**) Collector vessels LECs are of a cuboidal shape, connected by continuous zipper junctions (**D**) and surrounded by basement membrane and smooth muscle cells (grey), with intraluminal valves to aid passive transit of migratory immune cells (**E**).

**Figure 3 ijms-22-04458-f003:**
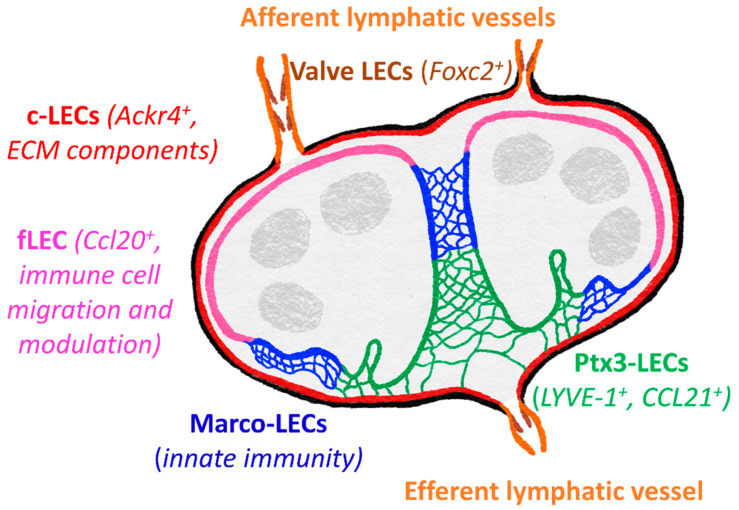
Subsets of lymphatic vasculature within the lymph node. Specialized gene expression profiles revealed five major distinct LEC subsets in both mouse and human lymph nodes [136]. These include valve cells of the afferent and efferent lymphatic vessels, SCS ceiling LECs (c-LECs) and floor LECs (f-LECs), and medullary sinus Marco-LECs and Ptx3-LECs. Genes of the c-LECs and Ptx3-LECs suggest roles in maintaining the structure of the lymph node, whereas f-LEC appear specialized for regulating lymph-borne immune cell entry and antigen presentation, with Marco-LECs important in innate immunity and response to pathogens. Ptx3-LECs also express numerous genes to suggest responsiveness to inflammation-induced remodeling and expansion of medullary and paracortical sinuses.

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
