# Peer review of "In Sickness and in Health: The Immunological Roles of the Lymphatic System"

_ijms, 2021, doi:10.3390/ijms22094458_

Round 1
Reviewer 1 Report
The author presents an extensive and well-presented overview of the current understanding of the lymphatic system in immunity and in immune regulation, with a focus on the lymphatic vasculature.
The review is very dense in information including also references to the very first studies of how immune cells traffic from tissues back to the venous circulation. It provides a very good base for any reader who wants to get an introduction into the immunological roles of the lymphatic system and how lymphatic vessels contribute to this, in homeostasis and in inflammation. The review also gives historical insight into the progression of this field with a very high number of references.
Minor edits:
Line 56-65: “Ovine models are particularly useful for studying lymphocyte traffic, as (in addition to ease of surgical access to individual lymph nodes) there is usually only a single efferent lymphatic vessel, draining each lymph node directly to the thoracic duct [4]. In humans and rodents, the “plumbing” is more complex, with lymph nodes often occurring in chains whereby the efferent lymph of one node contributes to the afferent supply of a second node. Thus, it is possible to examine the specificity of lymphocyte recirculation through normal tissue and under pathological conditions for extended periods of time. Such experiments demonstrated preferential re-circulation of distinct pools of lymphocytes, either through subcutaneous lymph nodes, mesenteric lymph nodes or through skin [5, 6].”
This paragraph is a bit difficult to follow. The sentence marked in cursive, as it is written, refers to the comment on human and rodents, but the next sentence “Such experiments...” reference 6 is from sheep. The way it is written also makes it sound like the “the anatomy of mouse and human” would make it easier to follow recirculation through “extended periods of time”, but it is not clear why and is probably not what the author intended to say.
The text in this paragraph should be re-structured to be easier to follow and understand.
Line 125: "Curiously, chemokines are not always simply synthesized by HECs and secreted or presented apically, like CCL21."
This statement is only true for mouse HEC. CCL21 is not expressed by human HECs, instead, they rely on FRC-derived CCL21 in a similar manner as the author suggests for CXCL12 and CXCL13. Although key HEV markers, like PNAd, is shared human and mouse, there are also known differences and the transcriptional profile of human HECs is still to be published.
Ref: 10.1182/blood-2004-11-4353. Carlsen et al.
The sentence should be modified.
Line 186: typo, write out what VE in VE-cadherin stands for.
Line 213: LEC produced Heparan Sulphate was recently shown to be dispensable for CCL21 presentation on dermal LECs. https://doi.org/10.3389/fimmu.2021.630002. Would be recommended to add a comment about this.
Line 271-72: “Of note, PLVAP is absent from peripheral LECs and was originally believed to be specific for blood vessel endothelium [90, 91].”
This statement is not completely correct. PLVAP is much lower expressed in peripheral LECs but it is not absent. Table 2 in ref. https://doi.org/10.3389/fimmu.2019.00816,
Section 5.1: For the comparison of human and mouse LN LECs the reference to 93 and 94 is sufficient (line 279). When the functions of the different subsets are discussed further in the text, a reference to https://doi.org/10.1371/journal.pbio.3000704 could also be included. This paper identified the same basic mouse LN LEC populations as described for mouse LN LECs in reference 93, but the authors do not name them in the same way and do not cross-map to human LN LECs.
Line 291-298: Add references to the statements.
Line 304: “(although Ptx3 itself is not expressed in peripheral LECs)”
Ptx3 is expressed in capillary vessels to different degrees in different organs of the mouse. ref. dataset GSE99743.
To be correct it is recommended that the sentence is modified slightly: “although Ptx3 is not reported to be highly expressed in mouse dermal LECs” or should be removed.
Line 565: The author here discusses the role of ACKR4 in LECs and skin keratinocytes. For the readers' information, it could be added, either here or in section 5.1, that the major demonstrated function of ACKR4 expression in LEC is the regulation of CCR7+ DC entry into the LN from the LN SCS, connected to the expression of ACKR4 in LN cLECs. Ref: doi: 10.1038/ni.2889. Epub 2014 May 11.
Line 698: “Primary dermal LECs secrete the monocytic and T-cell chemoattractants CCL2, CCL5, and CCL20, as well as the neutrophil-chemoattractants CXCL2, CXCL5 and CXCL8, predominantly from the luminal surfaces and thus it is highly likely that peripheral LECs are the sources of these chemokines in draining lymph nodes [140, 162].”
LN LECs are the main sources of some of these chemokines in homeostasis (e.g. CCL20 in LN fLECs, which is not expressed by mouse dermal LECs in homeostasis). Under inflamed conditions, the expression is further increased in LN LECs and FRCs of several of these chemokines (refs below). It is therefore highly unlikely that peripheral LECs are the only or main sources of these chemokines in draining lymph nodes in inflammation. The sentence should therefore be changed.
Some additional references in regards to inflammatory responses in LN LEC would also be recommended to be included, considering that section 10 is set out to specifically discuss the LN response in inflammation.
Refs inflammatory responses of LN LECs and stromal cells:
https://doi: 10.1038/ni.2262; (not included in the review)
https://doi.org/10.1016/j.celrep.2016.12.038 (not included in the review)
https://doi.org/10.3389/fcvm.2020.00052 ref 93
Line 794: Multiple cortical sinuses are situated adjacent to HEVs and under steady-state conditions, some lymphocytes access these sinuses within minutes of entering the lymph node and exit swiftly.
Add reference for this sentence. e.g. 97/102
Line 827: The author may in general consider indicating where references refer to studies based on genetic tools and specific targeting of LECs or are derived from antibody-based inhibition of pathways, the latter which limit the conclusions if the effect is only LEC dependent. E.g. ref 253.
Illustrations
The illustration of the peripheral and lymph node LECs are well designed and relevant.
Figure 4 legend: Add reference to ref 93, which the image is based on.
Considering the very dense information and the focus on LEC and immune cell interactions, the author could consider including a table (or two: homeostasis versus inflammation) with the key identified pathways for leucocyte and lymphocyte entry into peripheral initial capillary lymphatic vessels versus exit from the LN with respective references added. This would help the reader to get a better overview and would be highly valuable. This is however only a recommendation that would further improve an already very good review.
Reviewer 2 Report
This is a well-written comprehensive review, but there is a lack of focus in some chapters. If I understand correctly, the topic of this article is immunological roles of lymphatic vasculature in health and diseases. Some chapters describe about lymphocyte recirculation through high endothelial venules, interactions between DCs and FRCs/HEVs, and so on. These chapters should be re-written to align their focus with the main topic of the article.
Headings and sub-headings: Authors should re-consider the titles of headings. They are not well organized currently. 1. Introduction, 2. In a healthy steady state, 3. Entering lymph nodes from the blood, 4. Egress from peripheral tissues….. I believe chapter 3 and 4 also describe about a healthy steady state.
Figure 1 and 2: You may combine figure 1 and 2 since these figures do not have novel insights.
Chapter 5: You may add these references. Ulvmar MH, Nat Immunol 2014; Martens R, Nat Commun 2020.
Chapter 10.1: Xiang M et al used Oxazolone as LN inflammation model (Frontiers in Cardiovascular Medicine, 2020). You may clearly mention about this.
Chapter 11.2: You may cite this article regarding DC death: Tomura M, Scientific reports, 2014.
Figure 5 or table 1: It would be great if you could provide a new figure and/or a table regarding molecular mechanisms that regulate cell trafficking of various immune cells in the peripheral or lymph node lymphatics (e.g. LYVE1-HA-CD44 for DCs, CX3CL1-CX3CR1 for DCs, CCL21-CCR7 for DCs, T cells and neutrophils, Lewis X-CD209 for neutrophils)
Round 2
Reviewer 1 Report
The author has done all required changes and I recommend the article for publication in its present form.
Reviewer 2 Report
The authors have adequately addressed my concerns in their revised manuscript.